# Beyond Fish Oil Supplementation: The Effects of Alternative Plant Sources of Omega-3 Polyunsaturated Fatty Acids upon Lipid Indexes and Cardiometabolic Biomarkers—An Overview

**DOI:** 10.3390/nu12103159

**Published:** 2020-10-16

**Authors:** Heitor O. Santos, James C. Price, Allain A. Bueno

**Affiliations:** 1School of Medicine, Federal University of Uberlandia (UFU), Uberlandia 38408-100, Brazil; 2College of Health, Life and Environmental Sciences, University of Worcester, Worcester WR2 6AJ, UK; prij2_16@uni.worc.ac.uk (J.C.P.); a.bueno@worc.ac.uk (A.A.B.)

**Keywords:** alpha-linolenic acid, flaxseed, lipids, omega-3, walnuts

## Abstract

Cardiovascular diseases remain a global challenge, and lipid-associated biomarkers can predict cardiovascular events. Extensive research on cardiovascular benefits of omega-3 polyunsaturated fatty acids (n3-PUFAs) is geared towards fish oil supplementation and fish-rich diets. Nevertheless, vegetarianism and veganism are becoming more popular across all segments of society, due to reasons as varied as personal, ethical and religious values, individual preferences and environment-related principles, amongst others. Due to the essentiality of PUFAs, plant sources of n3-PUFAs warrant further consideration. In this review, we have critically appraised the efficacy of plant-derived n3-PUFAs from foodstuffs and supplements upon lipid profile and selected cardiometabolic markers. Walnuts and flaxseed are the most common plant sources of n3-PUFAs, mainly alpha-linolenic acid (ALA), and feature the strongest scientific rationale for applicability into clinical practice. Furthermore, walnuts and flaxseed are sources of fibre, potassium, magnesium, and non-essential substances, including polyphenols and sterols, which in conjunction are known to ameliorate cardiovascular metabolism. ALA levels in rapeseed and soybean oils are only slight when compared to flaxseed oil. *Spirulina* and *Chlorella*, biomasses of cyanobacteria and green algae, are important sources of n3-PUFAs; however, their benefits upon cardiometabolic markers are plausibly driven by their antioxidant potential combined with their n3-PUFA content. In humans, ALA is not sufficiently bioconverted into eicosapentaenoic and docosahexaenoic acids. However, evidence suggests that plant sources of ALA are associated with favourable cardiometabolic status. ALA supplementation, or increased consumption of ALA-rich foodstuffs, combined with reduced omega-6 (n6) PUFAs intake, could improve the n3/n6 ratio and improve cardiometabolic and lipid profile.

## 1. Introduction

Cardiovascular diseases remain a major Public Health concern, with lipid-associated biomarkers being trusted predictors of major cardiovascular events [1,2]. Added to the pharmacological strategies and conscious effort to improve the patients’ lifestyle by adequate nutrition and physical activity, nutraceutical strategies which include certain supplements, herbal extracts and functional foods, may be a helpful complementary approach for individuals with cardiovascular disease and dyslipidaemia [3,4,5,6,7,8,9].

The dietary replacement of saturated fats for polyunsaturated fatty acids (PUFAs) has shown beneficial effects upon lipid profile [10], as well as long-term protective benefits against major cardiovascular events and associated clinical complications [11,12]. For decades, omega-3 polyunsaturated fatty acids (n3-PUFAs) from either marine sources or fish oil (FO) supplementation were broadly referred to in cardiology guidelines [13,14,15]. For example, amongst several clinical investigations, Sagara et al. in 2011 showed that 2 g of DHA daily for five weeks improved blood pressure and lipid profile in a sample population of 38 middle-age men with hypertension and hypercholesterolaemia [16].

On the other hand, however, Manger et al. in 2010 did not find a significant trend in reduced risk of coronary events with increased consumption of n3-PUFAs from fish and fish supplements. Nonetheless, Manger argues that their sample population had a high intake of n3-PUA to begin with, and possibly the lack of association could have been attributed to a ceiling effect of n3-PUFAs [17]. Interestingly, a robust meta-analysis published in 2018, which included 79 Randomized Control Trials (RCTs) and a total of 112,059 participants, showed that n3-PUFAs supplementation actually did not show significantly greater efficacy on reducing the occurrence of cardiovascular events [18]. Such results must be interpreted carefully due to factors such as methodology employed, varied sample population investigated, other factors beyond the scope of the study, and a potential risk for bias. At the same time, it has also been demonstrated that plant sources of n3-PUFAs have shown some potential against cardiovascular events and dyslipidaemia [18].

The findings on fish consumption and FO supplementation upon cardiovascular health do not disregard the merit of investigating the usefulness of plant-derived n3-PUFAs in clinical practice. Studies on the potential efficacy of plant-derived n3-PUFAs become further justified as dramatic reductions of fish stocks have been reported in the North Atlantic Ocean and Mediterranean Sea [19,20]. In the present review, we have critically appraised the efficacy of plant-derived n3-PUFAs from foodstuffs, as well as its supplementation, upon the modulation of lipid profile and selected cardiometabolic markers. More specifically, we searched for the effects of chia seeds, flaxseeds, walnuts, as well as *Spirulina* and *Chlorella*. As those foodstuffs are gaining more popularity, a phenomenon possibly attributed to the increased awareness of environmental issues related to overfishing, combined with the increasing trend towards veganism, vegetarianism and flexitarianism across all segments of society, it is of greatest interest to clarify the clinical potential of plant sources of n3-PUFAs.

## 2. Methods

We employed the electronic databases Pubmed/Medline and Google Scholar to identify relevant publications. Randomised Controlled Trials (RCTs) written in English were the chosen sources of results as a means of translating current research findings into clinical practice. Preferred Reporting Items for Systematic Reviews and Meta-Analyses (PRISMA) guidelines [21] were used to evaluate and select the RCTs.

Limited to RCTs, we searched for plant sources of ALA by using the following combinations of Medical Subject Heading (MeSH) keywords: (“Chia Seeds” OR “Flaxseeds” OR “Hemp Seeds” OR “Walnuts” OR “Seaweeds” OR “*Spirulina*” OR “*Chlorella*”) AND (“LDL-C” OR “Low-Density Lipoprotein Cholesterol” OR “Total Cholesterol” OR “Triglycerides” OR “HDL-C” OR “Low-Density Lipoprotein Cholesterol” OR “Blood Pressure” OR “Cardiovascular Disease” OR “Cardiometabolic Risk” OR “Inflammatory Biomarkers” OR “Proinflammatory Cytokines”). The period covered in the search included inception to August 2020. Summary findings from the selected papers are presented in Appendix A. After perusal of such findings, we discussed key aspects and manually expanded the review by selecting further articles that have investigated nutrition facts, mechanisms of action and clinical applications.

As the number and robustness of clinical studies that have investigated the metabolic effects of plant-derived n3-PUFAs are only a fraction of those that employed FO, we have also summarized in Appendix A the outcomes of 38 selected FO supplementation studies identified using the key terms listed above and published in the last five years, with a combined sample population of 4136 individuals. The evidence gathered confirms that whilst some studies did show improvements in metabolic biomarkers after FO supplementation, some others did not.

## 3. Metabolic Pathways

### 3.1. Conversion of ALA in Humans

Not only the insufficient intake of dietary n3-PUFAs, but also the inefficient elongation and desaturation of ALA to eicosapentaenoic acid (EPA) and docosahexaenoic acid (DHA) in humans, result in low n3 long chain-PUFA content in blood and other peripheral tissues [22,23]. Accordingly, North America, Central and South America, and Africa, are geographical examples where EPA and DHA concentrations are endemically low [24]. More specifically, the intake of n3-PUFAs in the average USA population is low or very low [14].

Approximately only 5% of ALA is converted to EPA whilst less than 0.5% is converted to DHA in humans, although mammals have the essential enzymes used in this pathway [25,26,27]. A proof-of-concept study published in 2010 showed in newborn infants that only approximately 0.04% of ALA administered was elongated and desaturated to circulating EPA, whilst the subsequent conversion of EPA to DHA was comparatively more efficient [28]. The very low ALA to EPA bioconversion ratio identified in that study does not account for the EPA that was actually incorporated into solid tissues, but it does confirm the suggestion that even large amounts of dietary ALA will probably have negligible effects on plasma DHA levels [27].

Moreover, the conversion of ALA to EPA and subsequently DHA appears to be more efficient in women than in men, a phenomenon that could probably be explained by a possibly advantageous action of oestrogens in protecting the mother and the lactating child. Gender difference is a factor to be considered before making dietary recommendations for n3-PUFAs intake [25].

### 3.2. n3-PUFAs Versus n6-PUFAs

The inefficient bioconversion of ALA to EPA and DHA becomes more evident when considering that n3 and omega-6 (n6) PUFAs, although in much different concentrations in the typical westernized diet, compete almost equally for the same enzymatic pathway that elongates and desaturates the precursors ALA and linoleic acid (LA) to EPA and arachidonic acid (AA), respectively [29]. On the other hand however, potential roles for ALA in human health that may be independent of its bioconversion onto DHA have been proposed [27].

Experimental evidence alludes to a slower enzymatic metabolism of n3-PUFAs in relation to n6-PUFAs [30]. Therefore, focus on ALA intake is paramount for human health, but similarly important is the adequate intake of LA, as not to imbalance the n6/n3 ratio. Typical westernized diets show abundance of meats and poultry alongside deep-fried foodstuffs, an important dietary characteristic that favours high LA intake.

As seen in Figure 1, dietary sources of n6-PUFAs lead to raised levels of AA, culminating in increased synthesis of pro-inflammatory eicosanoids. Dietary sources of n3-PUFAs, in turn, allow the synthesis of anti-inflammatory eicosanoids. Increased n3-PUFA concomitant to decreased n6-PUFA decreases the AA content in platelets, vascular endothelial cells and vascular wall macrophages, thus reducing AA-derived pro-inflammatory mediators [31]. Cyclooxygenases (COX) and lipoxygenases (LOX) convert AA to prostaglandin E2, thromboxane A2, prostaglandin I2 and leukotriene B4, amongst other pro-inflammatory eicosanoids, whilst the same enzymatic pathways convert EPA to prostaglandin E3, thromboxane A3, prostaglandin I3 and leukotriene B5, amongst other anti-inflammatory eicosanoids [32,33,34].

### 3.3. Cardiometabolic Pathways

It has been observed in vitro that ALA is associated with decreased expression levels of vascular cell adhesion molecule-1 (VCAM-1), interleukin 6 (IL-6), proliferating cell nuclear antigen (PCNA), macrophage marker M3/84 (mac-3) and stearoyl-CoA desaturase-1 (SCD-1) [40]. Indirect effects of ALA upon cardiometabolic pathways have also been observed; ALA bioconversion to DPA is associated with decreased expression levels of COX-1, COX-2 and tumour necrosis factor-alpha (TNF-α), whilst its bioconversion to EPA and DHA is associated with decreased expression levels of peroxisome proliferator-activated receptor gamma (PPAR-γ) [40].

Given that such results have also been identified in aortic tissue [41], it is reasonable to speculate that the positive effects of ALA are driven by attenuation of inflammation, cell proliferation and oxidation [40]. In a more optimistic, long-term speculative scenario, such ALA-induced effects could retard the progression, or even promote an amelioration, of the atherosclerotic state.

ALA administration to primary cultured endothelial cells induced inhibitions of NAD-dependent deacetylase sirtuin-3 (SIRT3) reduction, superoxide dismutase 2 (SOD2) hyperacetylation, and mitochondrial reactive oxygen species (ROS) overproduction, alongside restoration of autophagic flux under damage induced by treatment with angiotensin II plus TNFα [42,43]. Such effects, apparently attributed to ALA, are in line with mitigation of endothelial dysfunction and experimental hypertension [42].

It is known that n3-PUFAs have the capacity to decrease liver triglyceride (TG) synthesis by competitive inhibition of 1,2 diglyceride acyltransferase (DAT), at the same time suppressing the activity of sterol regulatory element-binding protein 1c (SREBP-1c)—A protein that regulates the expression of genes involved in fatty acid and TG synthesis—and also to increase β-oxidation in adipose tissue [44,45]. Regarding the latter, the high affinity of n3-PUFAs for peroxisome proliferator-activated receptor alpha (PPAR-α) leads to a greater synthesis of enzymes involved in lipid catabolism, thus favoring not only fatty acid β-oxidation in peripheral tissues but also catabolism of circulating TG in chylomicrons and very low-density lipoprotein cholesterol (VLDL-C) [44,46,47]. Moreover, substrates for TG synthesis are also decreased by reduced transport of non-esterified fatty acids to hepatocytes, consequently reducing VLDL-C synthesis [44,48]. Nonetheless, as the majority of the cardiometabolic pathways so far elucidated are based on experimental data [40,41,42,43], a more thorough, critical and applied appraisal of the effects of ALA is needed before any general conclusions can be made.

## 4. Alternative Plant Sources of n3-PUFAs

### 4.1. Nuts and Seeds

#### 4.1.1. Nutrition Facts

Nuts and seeds are important sources of ALA and also micronutrients, polyphenolic compounds, sterols and fibres, which are protective elements against the exacerbation of chronic diseases [49,50,51]. For instance, nuts and seeds that are sources of ALA also contain a considerable amount of calcium, magnesium, and potassium (Table 1), which are fundamental macrominerals for cardiovascular health, mainly in the management of hypertension [52,53,54,55,56]. Furthermore, nuts and seeds are a source of protein. Although at lower levels as compared to animal-derived protein in terms of needs for muscle hypertrophy, the consumption of nuts and seeds is inversely correlated with cardiovascular events and mortality, as demonstrated by recent studies [57,58].

Regarding the ALA content of nuts and seeds, one ounce (28 g) of flaxseed, chia seeds, hemp seed or walnuts exceeds the Adequate Intake (AIs) for ALA, which is 1.1 g/day for women and 1.6 g/day for men, and 1.4 g/day and 1.3 g/day during pregnancy and lactation, respectively [59,60]. In one ounce, there are 6.38 g (398% and 580% of AIs for men and women) of ALA in flaxseed, 4.99 g (175% and 254% of AIs for men and women) in chia seeds, 2.80 g (312% and 453% of AIs for men and women) in hemp seed, 2.54 g (159% and 230% of AIs for men and women) in walnuts (Table 1).

#### 4.1.2. Walnuts

Walnuts are an important source of ALA [61,62]. A 2-years follow-up study recruited 236 elderly subjects, segregated into two groups: a control group without nut consumption, and an intervention group in which 15% of the approximate daily energy intake consisted of walnuts, at approximately 30–60 g/day of walnuts [63]. The researchers found a reduction of 8.5 mmHg in the systolic blood pressure, whose baseline levels were >125 mmHg; however, no changes were observed in diastolic blood pressure. In the same study, the participants who consumed walnuts required lower dosages of antihypertensive drugs as compared to the control participants. The blood pressure of all participants in that study was monitored by the 24-h ambulatory blood pressure monitoring, which is considered the gold standard in the diagnosis of hypertension [64]. In contrast, a recent meta-analysis [65] did not support walnut consumption per se as a blood pressure-lowering strategy. Despite being a meta-analysis, that study [65] may have had a few limitations due to heterogeneity. The results of Domènech et al. [63] provide what appears to be reliable evidence, based mainly on its long-term duration and sample size.

Le et al. [66] recruited 213 overweight and obese women to a weight loss study, and offered one of the three following dietary regimens: a walnut-rich diet which consisted of 35% energy from fat and 45% energy from carbohydrates, or a low-fat (20%) high-carbohydrate (65%) diet, or a high-fat (35%) low-carbohydrate (45%) diet. After six months of intervention, high-density lipoprotein cholesterol (HDL-C) levels were significantly increased (*p* < 0.05) in the walnut-rich group (from ≈60 to ≈63 mg/dL), whilst a small decrease was observed in the low-fat (from ≈60 to ≈57 mg/dL) and low-carbohydrate (from ≈58 to ≈57 mg/dL) groups.

Interestingly, Fatahi et al. [67] randomised 99 overweight and obese women into three low energy-diet groups: the first group consisted of 300 g/week of fatty fish such as salmon, avoiding the intake of plant sources of n3-PUFAs; the second group consisted of 18 walnuts/week, avoiding the intake of fish and other plant sources of n3-PUFAs; the third group consisted of 150 g fatty fish and nine walnuts/week, avoiding the intake of other sources of n3-PUFAs. After 12 weeks of dietary intervention, as compared to the fish-only group and walnut-only group, the fish + walnut group showed better metabolic profile overall, evidenced by greater increase in HDL-C (+3.6 mg/dL) levels, followed with greater decrease in systolic blood pressure (−5 mmHg), fasting blood glucose (−12 mg/dL), low-density lipoprotein cholesterol (LDL-C) (−6 mg/dL), high-sensitivity C-reactive protein (hs-CRP) (−0.51 mg/L), D-dimer (−0.45 mg/dL), fibrinogen (−22 mg/dL), alanine aminotransferase (ALT) (−6 IU/L), aspartate aminotransferase (AST) (−6 IU/L), TNF-α (−0.08 ng/mL) and IL-6 (−1.6 ng/mL).

#### 4.1.3. Flaxseed

Flaxseed is an important source of ALA, and a few trials have identified beneficial effects of flaxseed intake upon lipid indexes and cardiometabolic biomarkers. In a study recruiting 21 patients with coronary artery disease [68], a pivotal population to ascertain the magnitude of a cardiometabolic intervention, the daily consumption of 30 g flaxseed for 12 weeks promoted better outcomes as compared to the control group in increasing flow-mediated dilation (5.1 vs. −0.55% change from baseline for the flaxseed and control groups, respectively), whilst decreasing the inflammatory status by reducing the levels of CRP (−1.18 mg/L), IL-6 (−7.65 pg/mL), and TNF-α (−34.73 pg/mL). Importantly, no significant change in body weight was observed in either groups [68], which appears to be a very relevant result as it suggests that flaxseed may improve cardiovascular parameters independently of weight loss.

In a systematic review and meta-analysis of RCTs with 1502 patients across 32 studies, flaxseed or its derivatives (whole or ground flaxseed, flaxseed oil, or lignan supplements) reduced the concentrations of hs-CRP (weighted mean difference (WMD): −0.75; 95% CI −1.19, −0.31) and TNF-α (WMD: −0.38; 95% CI −0.75, −0.01) but did not change IL-6 levels. Flaxseed was tested in the form of whole flaxseed, golden flaxseed, flaxseed oil, and lignan supplements at dosages ranging from 360 mg to 60 g, for 2 to 12 weeks, with an averaged intervention period of approximately 10 weeks [69].

A recently published clinical trial recruited 41 women suffering with polycystic ovary syndrome, randomly segregated into two groups, group 1 subjected to lifestyle changes (American Heart Association recommendations + >30 min moderate to intense activity 3x/week) plus 30 g/day brown flaxseed flour in salad, yogurt or cold drinks, and group 2 subjected to the same lifestyle changes only, for 12 weeks [70]. The authors found that the flaxseed group showed significant improvements in insulin, homeostasis model assessment of insulin resistance (HOMA-IR), TG, hs-CRP, IL-6, leptin, HDL-C and adiponectin, as compared to the non-flaxseed group [70].

In a RCT recruiting 100 eligible patients suffering with non-alcoholic fatty liver disease (NAFLD), Yari et al. [71] found that 30 g flaxseed daily plus positive lifestyle interventions for 12 weeks decreased serum concentrations of total cholesterol (TC) (−31.71 mg/dL), TG (−61.33 mg/dL), LDL-C (−22.64 mg/dL), ALT (−11.12 U/L), AST (−5.37 U/L) and gamma-glutamyltransferase (−11.54 U/L), results that were not matched in the group submitted to positive lifestyle interventions only. It should be noted however that both groups showed reductions in BMI (30.37 ± 4.42 to 28.05 ± 3.89 kg/m^2^ in the flaxseed plus lifestyle improvement group, and 33.37 ± 5.56 to 32.42 ± 5.98 in the lifestyle improvement only group) as well as the intensity of hepatic steatosis, a result most likely attributed to decreased energy intake in both groups (2379.41 ± 473.74 to 2117.47 ± 378.46, and 2424.45 ± 470.89 to 1966.39 ± 449.52 kcal, respectively). Such findings reinforce the hypothesis of beneficial effects of flaxseed independently of changes in energy intake and body composition.

In a mirrored RCT [72], this time recruiting 98 patients suffering with metabolic syndrome, the same researchers from the previously mentioned study [71] observed comparable results, in which 30 g flaxseed plus positive lifestyle interventions for 12 weeks reduced by 76% the prevalence of metabolic syndrome, whilst the lifestyle intervention only group had this parameter reduced by 36.4% (*p* = 0.013 for the difference between groups). Likewise, both groups reduced their calorie intake before versus after, but without differences between them (2423.04 ± 468.98 to 2198.76 ± 455.47 and 2410.26 ± 451.87 to 2079.89 ± 465.46 for flaxseed and control groups, respectively).

### 4.2. Oils

#### 4.2.1. Lipid Profile of Oils

Oils rich in ALA are a powerful tool to investigate the effects of ALA within a less complex matrix. Despite the presence of other fatty acids and fat-soluble compounds in the oil, the removal of fibre, vitamins, especially water-soluble ones, and water-soluble matter surely minimize the effects of confounding variables. Flaxseed, walnut, and rapeseed oils are, respectively, the main sources of ALA. Soybean oil is often considered a small to reasonable source of ALA, once research into its fatty acid composition has shown ALA concentrations ranging from 2.7% to 7.8% [73,74]. The n6-PUFAs content is nonetheless crucial when contemplating the n3 content of plant oils, as sunflower, corn, walnut, cottonseed, soybean, and peanut oils are important sources of n6-PUFAs (Table 2).

#### 4.2.2. Flaxseed Oil

A RCT investigated the effects of either 25 mL/d flaxseed oil or 25 mL/d sunflower oil administered for seven weeks to 60 patients suffering with metabolic syndrome [75]. Serum IL-6 levels decreased significantly in both groups (9.37 to 7.90 pg/mL, *p* < 0.001 for flaxseed oil, and 9.22 to 8.48 pg/mL, *p* < 0.006 for sunflower oil), but the flaxseed oil group presented a greater reduction (*p* = 0.017). Given that that was a dosage considered high, it is worth mentioning that no side effects were reported in either group [75]. Interestingly, in a study recruiting 60 women with gestational diabetes [76], the daily supplementation for 6 weeks with 2 g/day flaxseed oil capsules, which contained 800 mg/day ALA, reduced the concentrations of TG (−40.5 mg/dL), TC (−22.7 mg/dL), insulin (−2.2 µIU/mL), and hs-CRP (−1.3 mg/L), as compared to a matched group that received sunflower oil capsules. The flaxseed oil-receiving group also showed upregulated LDL receptor, downregulated IL-1 and TNF-α gene expression, decreased malondialdehyde levels and increased total nitrite and total glutathione levels.

In a recent study [77] recruiting 59 overweight and obese adults with stage I hypertension without pharmacological treatment, 10 g of refined cold-pressed flaxseed oil (4.7 g ALA) for 12 weeks decreased fasting free fatty acid (−58 μmol/L) and TNF-α (−0.14 pg/mL) plasma concentrations. In contrast, no changes were found in other metabolic risk markers (e.g., serum glucose and TG levels) nor vascular function markers (e.g., brachial artery flow-mediated vasodilation, carotid-to-femoral pulse wave velocity, and retinal microvascular calibres) before versus after testing, both on fasting and postprandially. We consider this result to be extremely relevant for our critical discussion, as the ALA intake in that study was about three to five times higher than the recommended daily intake and, even so, it failed to improve cardiovascular markers. Furthermore, the volunteers of that study not only had obesity or overweight and stage I hypertension, their average age was 60 ± 8 years, a finding that is positively associated with vascular ageing, which by definition poses a greater risk for hypertension and atherosclerotic disease than the more traditional risk factors, including lipid and glucose levels, smoking and sedentary lifestyle [78].

The Omega-3 Index (O3I) reflects the relative percentage amount of EPA and DHA in erythrocyte membranes, and is considered a surrogate biomarker for cardiovascular events [79]. Cao et al. investigated in individuals with low baseline n3-PUFA levels the effects of supplementation with fish oil or flaxseed oil upon O3I [80]. Cao found that supplementing 2.1 g/day FO (1296 mg EPA + 864 mg DHA) for eight weeks increased the O3I from 4.3% to 7.8% (*p* < 0.001), followed by a gradual decline to 5.7% and to 3.8% at 4 and 16 weeks after the end of the supplementation period, respectively. On the other hand, supplementation with flaxseed oil (3510 mg ALA + 900 mg LA/d) for the same period, in turn, did not significantly change the O3I, but it did increase not only EPA but also n3-docosapentaenoic acid (DPA) [80], a fatty acid that sits in between EPA and DHA in the elongation desaturation pathway.

It can be argued that the study of Cao et al. [80] did not cover a period of supplementation that would allow maximum incorporation and saturation of supplemented PUFAs into erythrocyte membranes, therefore not raising the O3I to its maximum achievable level. A O3I higher than 8% has been proposed to be favourable against cardiovascular events, whilst ≤ 4% is interpreted as low [24,79,81]. Accordingly, the benefits of ALA as a mitigator of cardiometabolic events can be supported by its role as a substrate for EPA. The latter is a well-known element against pro-inflammatory pathways in the cardiovascular system.

#### 4.2.3. Soybean Oil

Despite having an obviously different fatty acid profile as compared to FO, soybean oil, unarguably, is a source of some ALA, and a few studies have already demonstrated a mild but relatively positive potential for soybean oil. For example, in treated hepatitis C patients (*n* = 52), 6000 mg/day FO or a soybean oil control for 12 weeks significantly decreased serum insulin levels (17.1 to 10.9 µIU/mL *p* = 0.001, 12.6 to 10.6 µIU/mL *p* = 0.011 for FO and soybean oil groups, respectively) and HOMA values (3.8 to 2.4 *p* = 0.002, 3.1 to 3.0 *p* = 0.046 for FO and soybean oil groups, respectively) when comparing baseline versus end-of-intervention. The FO group was clearly more efficient; however, differences between both groups (*p* = 0.016 for insulin levels and *p* = 0.015 for HOMA-IR values) have been observed [82].

In a small controlled clinical trial recruiting 16 hypercholesterolaemic women in a 10 weeks intervention, participants received an amount of soybean oil that consisted of 20% of their energy intake in a weight-maintaining diet with <300 mg/day of cholesterol [83]. The control group consisted of the same participants in a weight-maintaining diet with <300 mg/day of cholesterol for eight weeks but no soybean oil. As compared to the eight-week control period, TC, HDL-C, LDL-C and small dense low-density lipoprotein-cholesterol (sdLDL-C) levels were reduced at the end of the 10 week-soybean oil intervention period [83]. Furthermore, the sdLDL oxidation lag time was reduced after soybean oil consumption. In addition to the unfavourable cardiovascular conditions of maintaining low HDL-C levels, the former is linked to the pathophysiology of atherosclerotic disease due to the damage potential of sdLDL-C subclasses, especially in their oxidised form, on arterial structures [6].

#### 4.2.4. Rapeseed Oil

In a meta-analysis of 27 RCTs [84], consumption of rapeseed oil was associated with reductions of approximately 7.24 mg/dL in TC (95% CI, −12.1 to −2.7) and of approximately 6.4 mg/dL in LDL-C (95% CI, −10.8 to −2) serum levels, as compared to sunflower oil and saturated fat. No changes were observed in TG nor HDL-C. Overall, the daily dose of rapeseed oil ranged from 12 to 50 g for 21 to 180 days. Most trials in that meta-analysis addressed individuals with lipid disorders and patients with heart disease, type 2 diabetes mellitus, obesity, metabolic syndrome, or non-alcoholic fatty liver disease. Importantly, the trials were under isocaloric conditions and, thus, partially avoided the bias of weight loss and its relationship with amelioration of lipid indices [84]. A clinical trial study in well-controlled conditions, recruiting 10 healthy men aged 25.3 ± 1 years, found that 24 h lipid oxidation was more pronounced when the participants received rapeseed oil-enriched meals for the duration of the study, as compared to a matched meal enriched with palm oil, a source of saturated fat [85].

It is well known that replacing dietary saturated fats with monounsaturated fatty acids (MUFAs) and PUFAs can reduce overall mortality [86]. The United Kingdom Scientific Advisory Committee on Nutrition in 2019 concluded that the recommendation for the population average contribution of saturated fat to total calorie intake should remain at no more than 10% of total dietary intake, and that reducing intakes of saturated fats reduces the risk of cardiovascular and heart disease events [87].

The alleged cholesterol-lowering properties of rapeseed oil could be attributed not only to its ALA content, but also possibly combined with its MUFA composition. Approximately 56% of the total fatty acids in rapeseed oil are MUFAs, with oleic acid as the most abundant one, at 54.5% of the total fatty acids, approximately [88]. The abundance of oleic acid in rapeseed oil may support its beneficial properties, as it has been shown that oleic acid elicits improvement on lipids and lipoproteins, as well as reduced risk of cardiovascular disease in humans [89]. The ALA content in rapeseed oil, in turn, is estimated at approximately 6 to 10% of total fatty acids [59,90,91]. Interestingly however, a study found a much lower ALA content in rapeseed oil, ≈1.2% [92].

A critical interpretation of published studies will see that other plant sources of MUFAs and PUFAs may show positive results in comparison to rapeseed oil. For instance, the consumption of sesame oil was more favourable for glycaemic control markers when compared to rapeseed oil in a recent RCT recruiting individuals with type 2 diabetes mellitus [91]. In that study, rapeseed oil increased serum fasting blood glucose (+7.72 ± 3.15 mg/dL, *p* < 0.05), whilst sesame oil decreased serum insulin (−6.00 ± 1.72 mIU/mL, *p* < 0.05) levels in a nine-week intervention period. The dietary recommendation was based on 30–32% of the total energy intake from fats but, despite the predominance of oil intake in each intervention, we noticed that the authors did not present in their study the exact or estimated amount of oil consumed. Some of the strengths of the study include its design, a triple-blind, cross-over clinical trial with 92 subjects completing all treatment periods, which were composed of four-week run-in and four-week wash-out periods based on sunflower oil.

### 4.3. Seaweed

#### 4.3.1. Lipid Profile of Seaweed

The use of seaweed for cooking and as a food supplement is gaining more popularity worldwide [93,94]. As a functional food, seaweed is a vegetarian source of n3-PUFAs, protein, and micronutrients [95]. Spirulina and chlorella are commercially available biomass extracts of cyanobacteria and green algae respectively, designed to attend a growing demand [96]. Both spirulina and chlorella are often claimed to be valuable sources of n3-PUFAs; however, the accuracy of such statement needs to be clarified in context, as to obtain approximately 2 to 3 g of total lipids from these microalgae it is necessary to ingest approximately 28 g of it in its powdered form. The total lipid content can be practically zero in supplemental dosages (≈3 g/day) [59].

*Chlorella minutissima* UTEX 2219 and UTEX 2341 feature 3.3% and 31.3% EPA, respectively, of the total fatty acid content, but DHA was not detected in either strain [97]. In a fatty acid profile analysis of *Spirulina platensis* from seven commercially available products, EPA and DHA were detected in only two samples, contributing with 1.79% and 7.70%, and 2.28% and 2.88%, respectively, of the total fatty acids [98]. Similarly to flaxseed and nuts, whether the effects of dietary intervention with seaweed are attributed to its n3-PUFAs per se remain to be investigated, as the food matrix should be considered. *Spirulina* and *Chlorella* contain not only macro and micronutrients but also other compounds with antioxidant properties which may play a role in positive health outcomes [99,100,101].

#### 4.3.2. Clinical Findings

A meta-analysis published in 2016 [102] found that in general the daily consumption of 1 to 10 g *Spirulina* led to significant improvements in lipid profile by reducing TC in ≈47 mg/dL (95% CI: −67.31 to −26.22), LDL-C in ≈41 mg/dL (95% CI: −60.62 to −22.03), TG in ≈44 mg/dL (95% CI: −50.22 to −38.24), whilst increasing HDL-C in ≈6 mg/dL (95% CI: 2.37–9.76). Seven placebo-controlled clinical trials with duration of 2–4 months were included in that meta-analysis, and the population appraised consisted of patients with diabetes, cardiac diseases, nephrotic syndrome and HIV infection, illnesses whose pathophysiologies are related to dyslipidaemia [103,104,105]. Seemingly, an effective and practical dose was about 4 g/day of *Spirulina*, which can be administered in capsules or powder [102]. Nevertheless, as discussed above, we believe that the n3-PUFAs content in *Spirulina* was not the sole player in yielding such outcomes.

Supplementation with 300 mg/day of *Chlorella* for 8 weeks decreased TNF-α levels, in comparison with the placebo group, in a RCT of 70 patients with NAFLD [106]. An 8 week-long RCT investigating 44 women suffering with primary dysmenorrhea and supplemented with 1500 mg/day *Chlorella* found a significant reduction in hs-CRP levels (from 2590.00 ± 1801.66 to 974.21 ± 292.85 ng/mL), as compared to the control group [107].

In a Japanese sample population, 40 daily tablets of *Chlorella* were provided to 17 individuals with borderline high fasting blood glucose, TC, and TG levels, as well as to 17 healthy individuals [108]. After 16 weeks of supplementation, the researchers found reductions in body fat percentage, serum TC, and fasting blood glucose levels in both groups [108]. In our view however, that study appears to show a few limitations that should be considered in the context of a broader clinical scenario. The researchers did not measure food intake, there was no matched control group, and body fat percentage was obtained through bioelectrical impedance, which as a method of body composition analysis has some limitations [109]. Lastly, *Chlorella* supplementation was used in an impracticable dosage that, although it may be tested, cannot be translated into a broader clinical recommendation. As 40 tablets were ingested together with a considerable volume of fluid every day, probably this posology in itself with fluid may have resulted in lower food intake due to stomach filling.

## 5. Population

Type 2 diabetes, obesity, dyslipidaemia and hypertension, conditions that we have attempted to address in the present study, are related to a pro-inflammatory state [110,111,112,113]. The evidence so far available suggests that the consumption of ALA-rich foodstuffs may attenuate the levels of inflammation-associated biomarkers. More importantly however, the consumption of ALA food sources appears to be associated with reduced incidence of cardiovascular events. Further long-term RCTs are imperative to further elucidate such preliminary findings.

In the context of liver diseases, NAFLD particularly, lifestyle improvement is a cornerstone in ameliorating the disease. We have identified studies that showed beneficial effects of ALA interventions in patients with NAFLD [71,84]. Accordingly, a favourable nutritional support based on ALA food sources could be considered in therapies for NAFLD patients in cases of low frequency of fatty fish intake or absence of FO supplementation.

Food sources of ALA may also be relevant during pregnancy and lactation due not only to their rich nutritional composition but also due to a thoroughly justified need to avoid complex herbal supplement mixtures that may jeopardize the health of the mother and of the child. Additionally, plant sources of n3-PUFAs could be seen as an option for pregnant women who do not tolerate fatty fish and for those who suffer with nausea and with hyperemesis gravidarum, relatively common manifestations during the gestational period [114,115]. The amount and profile of n3-PUFAs ingested by the lactating mother is paramount for infant health, as the mother’s diet directly reflects upon her milk fatty acid profile [116].

Exclusively vegan diets are to be examined carefully due to the risk of n3-PUFAs deficiency. Apart from a lower intake of total and saturated fats, another characteristic of exclusively vegan diets is a higher proportional intake of n6-PUFAs, when compared to omnivorous and vegetarian diets [110,117]. For those reasons, recommendations for vegan diets that include appropriate amounts of ALA, necessarily combined with a balanced n3/n6 ratio, are paramount for the maintenance of long-term health.

It is widely accepted that peanut and peanut butter have been speculated by the layperson and by some health practitioners as friendly components of the exclusively vegan diet. Careful consideration however should be exercised regarding the amounts of peanut and peanut butter consumed, as a pilot study observed in 14 exercise-trained healthy individuals with an average age of 30 years that the daily consumption of approximately 103 g of peanut butter for 4 weeks led to changes in body composition, markedly increased body fat content [118]. Such results only confirm that careful calorie counting is pivotal when adding a new source of lipids to the diet plan, regardless of whether those are considered healthy food items. Furthermore, peanut is particularly rich in LA and virtually absent of ALA (Table 2).

## 6. Decision-Making Practice

A diet poor in n3-PUFAs and rich in n6-PUFAs in the long term leads to inflammation and increased risk of diseases, including cardiovascular diseases [119]. If the patient does not comply with a diet that includes fatty fish or FO supplements, ALA-rich foods are important to at least partially supply some of the n3 requirement, at the same time reducing the LA bioavailability. Furthermore, some individuals in society may choose not to adhere to food supplementation, be it because of added costs, dietary choices such as veganism and vegetarianism, the wish to avoid polypharmacy if they are already taking medicine tablets, or due to personal values. Such lifestyle choices and circumstances further emphasize the need for a careful nutritional planning that includes sources of n3-PUFAs.

Along those lines, some period supplementing FO and followed by some period incrementing plant sources of n3-PUFAs may be an example of what would be a ”periodized” loading of n3 status. Such hypothesis can be sustained due to the incorporation of n3-PUFAs into cell membranes, which occurs over a few months after supplementation, whilst the period appears to be longer for decreasing n3 status than the progress of storage. In the case of the study of Cao et al. [80] for example, two months of FO supplementation was sufficient to almost achieve a reasonable n3 state, which was maintained by one month afterwards but returning to the same baseline level after 16 weeks of interruption. We believe that a ”periodized” FO supplementation regimen could be conceivable two to three times a year for one to four months, followed with nutritional counselling and observing the intake of foodstuffs rich in ALA in the period without the FO supplementation, hoping to provide a better n3-DPA level. Nutritional advice however will always be tailored to the individual’s needs and preferences, such as frequency of consumption of fatty fish and food sources of ALA, patient’s income, and any coexisting morbidity. As nutritional strategy for individuals who, for one reason or another, do not eat fish nor take FO supplementation, adding in ALA-rich foods, e.g., flaxseed and chia seeds, or adding their oils within a dietary plan, alongside a careful consideration to dietary sources of n6-PUFAs, in itself seems to be paramount for long term overall health.

## 7. Perspectives

In the present study we focused on the effects of ALA found naturally in foodstuffs, seeds, nuts and oils, instead of isolated ALA supplementation. Further clinical research is urgently required to broaden the current knowledge of the potential of ALA upon cardiometabolic dysregulations and cardiometabolic protection. Well-designed RCTs could certainly minimize the residual confounding variables caused by other nutrients.

The limitations of our study, as well as of other studies that have investigated ALA cardiometabolic effects specifically, are many. Ultimately however, although the current study provides some insights in a real-world prescription-based intervention, we also encourage the execution of meta-analyses to expand the current knowledge of specific nutrients in specific populations, in an attempt to find ideal dose-responses, as well as to continuously update guidelines in nutrition and cardiology.

## 8. Conclusions

In case of suspected insufficient n3 status, such as in individuals with low intake of fatty fish, those who do not take FO supplement, and in vegan individuals with very narrow dietary habits, alternative plant sources of n3-PUFAs may be candidates for partially attending the n3 metabolic demands. Although plant sources of n3-PUFAs are less impactful on EPA and DHA levels, evidence suggests that those are foodstuffs positively associated with favourable cardiometabolic outcomes, which could be triggered by other plant components, in synergy with ALA. Not only ALA in isolation, but its proposed effect in combination with other plant fatty acids and other plant components such as fibre, potassium, magnesium, and non-essential substances, e.g., polyphenols and sterols, may be the players in yielding benefits in cardiovascular metabolism.

Consumption of walnuts and flaxseed seems to be the main plant sources of n3-PUFAs with strong scientific basis for translation into clinical practice. Regarding oil intake, we believe that flaxseed oil is more advantageous than walnut oil, because the former’s ALA content is five times greater than that of the latter, which in turn, can be considered the second principal source of ALA. Although several studies have alluded to rapeseed and soybean oils as ALA sources, their ALA amount is slight when compared to flaxseed oil, so that a usual oil serving must be considered in order not to exceed the daily energy requirement in an attempt to achieve an optimal level of ALA. Regarding seaweed, *Spirulina* and *Chlorella* have gained attention but there are no discernible studies corroborating a relevant amount of n3-PUFAs in usual doses of supplementation. Seemingly, the benefits of seaweed over cardiometabolic markers appear to be driven by their antioxidant content.

The introduction of ALA-rich foods is a cornerstone for individuals looking for n3 sources beyond fish and fish oil. It is nevertheless of greatest concern that the proposal to increase the consumption of ALA ought to be integrated with a controlled calorie intake and controlled n6-PUFAs intake, since both of them can be raised concomitantly, thus ensuing in untoward effects such as increased fat mass and cardiometabolic dysregulations.

## Figures and Tables

**Figure 1 nutrients-12-03159-f001:**
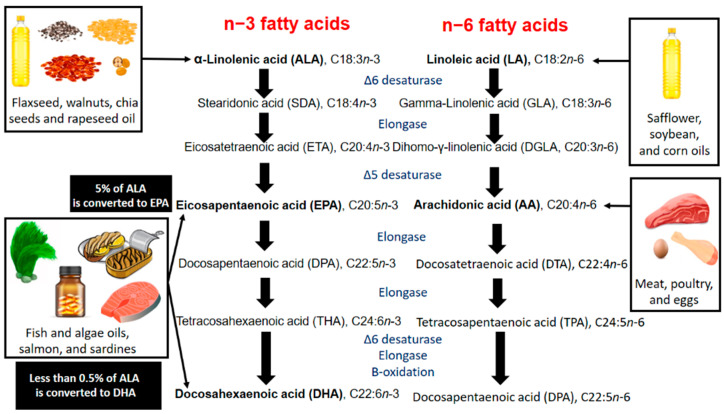
Whilst the main dietary sources of the n3-PUFAs EPA and DHA are fatty fish (e.g., salmon and sardines) and algae, the n3 precursor ALA is found mainly in walnuts, chia seeds, flaxseeds, and rapeseed oil. Likewise, meat, poultry and eggs are important sources of the n6-PUFAs AA, and its n6 precursor LA is found mainly in safflower, sunflower, soybean and corn oils [35,36,37]. ALA and LA are converted by desaturases and elongases to EPA and AA, respectively, and subsequently converted in a series of complex enzymatic reactions to the longer forms DHA and n6-docosapentaenoic (n6-DPA), respectively [38,39].

**Table 1 nutrients-12-03159-t001:** Nutrition facts of ALA-containing seeds.

Food Item, one Ounce /≈28 g [FDC ID]	Energy (kcal)	Protein (g)	Total Lipid (g)	Total Fiber (g)	CHO (g)	Ca (mg)	Mg (mg)	K (mg)	LA, 18:2, *n-*6 (g)	ALA, 18:3, *n-*3 (g)	*n-*6 /*n-*3 Ratio
Chia seeds [170554]	136	4.63	8.60	9.75	11.79	176.68	93.8	113.96	1.63	4.99	0.32
Hemp seed [170148]	155	8.83	13.65	1.12	2.42	19.6	196	366	7.68	2.80	2.74
Flaxseed [169414]	150	5.12	11.80	7.64	8.08	71.4	109.76	227.64	1.65	6.38	0.25
Walnuts [784410]	183.12	4.26	18.25	1.9	3.83	27.44	44.24	123.48	10.8	2.54	4.25

Adapted from the United States Department of Agriculture (USDA) database [59]. ALA, alpha-linolenic acid; CA, calcium; CHO, carbohydrate by difference; FDC, FoodData Central; K, potassium; LA, linoleic acid; Mg, Magnesium.

**Table 2 nutrients-12-03159-t002:** ALA content and principal nutrition facts of selected edible oils.

Oil Type, 1 Tablespoon/13.6 g [FDC ID]	Energy (kcal)	Total Lipid (g)	Vitamin E (mg)	Saturated Fats (g)	Monounsaturated Fats (g)	Polyunsaturated Fats (g)	LA, 18:2/*n**-*6 (g)	ALA, 18:3 *n**-*3 (g)	*n*-6/*n**-*3 Ratio
Flaxseed oil [789037]	120	13.6	0.064	1.22	2.51	9.23	1.95	7.26	0.26
Walnut oil [789048]	120	13.6	0.054	1.24	3.1	8.61	7.19	1.41	5.09
Rapeseed oil [172336]	120	13.6	2.38	1.01	8.64	3.82	2.58	1.24	2.08
Soybean oil [789045]	120	13.6	1.11	2.13	3.1	7.85	6.93	0.92	7.53
Corn oil [789035]	122	13.6	1.94	1.76	3.75	7.44	7.28	0.158	46.07
Olive oil [789038]	120	13.6	1.94	1.86	9.85	1.42	1.32	0.103	12.81
Cottonseed oil [789036]	120	13.6	4.8	3.52	2.42	7.06	7	0.027	259.25
Coconut oil [789034]	121	13.5	0.015	11.2	0.861	0.231	0.229	0.003	76.33
Peanut oil [789039]	120	13.6	2.12	2.28	6.24	4.32	4.32	0	-
Almond oil [789033]	120	13.6	5.33	1.12	9.51	2.37	2.37	0	-
Sunflower oil [789047]	120	13.6	5.59	1.4	2.65	8.94	8.94	0	-

Highest to lowest sources of ALA among common oils used for cooking. In addition, the main sources of LA can be noted, which are sunflower, corn, walnut, cottonseed, soybean, and peanut oils, respectively. Adapted from the United States Department of Agriculture (USDA) database [59].

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
