# Peer review of "Beyond Fish Oil Supplementation: The Effects of Alternative Plant Sources of Omega-3 Polyunsaturated Fatty Acids upon Lipid Indexes and Cardiometabolic Biomarkers—An Overview"

_nutrients, 2020, doi:10.3390/nu12103159_

Round 1
Reviewer 1 Report
The manuscript by Santos et al. “Beyond Fish Oil Supplementation: The Effects of Alternative Plant Sources of Omega-3 Polyunsaturated Fatty Acids upon Lipid Indexes and Cardiometabolic Biomarkers – An overview” reports about the health effects of alpha-linolenic acid (ALA) and plant products that.contain ALA.
The authors describe the literature search they performed, which focussed on natural porducts containing ALA, but it excluded studies looking at ALA supplements. This point should be discussed in the manuscript.
It is explained that conversion of ALA to long chained derivatives in humans is not the reason for positive effects, but it should be clearer outlined, what alternative mechanistic reasons could exisit.
As the introduction mentions other components than ALA of the products discussed (walnuts, flaxseed oil, rapeseed oil, seaweed) could be the responsible for the positive effects. One important point in this context, which is not really touched are other fatty acids with positive health effects (e.g. MUFA in rapeseed oil).
The authors mention individual studies, but they do not try a meta-analysis. This agrees with the title (an overview), but one would whish to know the reason described why this has not been attempted.
An important point in relation to fatty acid intake is the balance between n-6 and n-3 fatty acids. Could it be that some of the positive aspects ascribed to the ALA containing products could be due to reduced linoleic acid availability?
Specific points
Line 66: warrant instead of are warranted
Line 92: LC-PUFA instead of PUFAs
Figure 1: the pathway for n-6 PUFA seems not correct
Table 1: might be clearer to give the contents per 100g instead of per table spoon (13.6 g)
Line 302: an explanation would be good, how refinement could selectively eliminate ALA from rapeseed oil (considering that fatty acids are triglyceride bound)
Author Response
Reviewer 1
The manuscript by Santos et al. “Beyond Fish Oil Supplementation: The Effects of Alternative Plant Sources of Omega-3 Polyunsaturated Fatty Acids upon Lipid Indexes and Cardiometabolic Biomarkers – An overview” reports about the health effects of alpha-linolenic acid (ALA) and plant products that.contain ALA. The authors describe the literature search they performed, which focussed on natural porducts containing ALA, but it excluded studies looking at ALA supplements. This point should be discussed in the manuscript.
Response: Dear reviewer, thank you very much for your comments. We have discussed this point in the topic ‘Perspectives’.
It is explained that conversion of ALA to long chained derivatives in humans is not the reason for positive effects, but it should be clearer outlined, what alternative mechanistic reasons could exisit.
Response: We have added two subtopics “3.2. n3-PUFAs versus n6-PUFAs and 3.3. Cardiometabolic pathways) in order to discuss the alternative mechanistic reasons more clearly.
As the introduction mentions other components than ALA of the products discussed (walnuts, flaxseed oil, rapeseed oil, seaweed) could be the responsible for the positive effects. One important point in this context, which is not really touched are other fatty acids with positive health effects (e.g. MUFA in rapeseed oil).
Response: In the subtopic “4.2.4. Rapeseed oil” we have touched upon MUFA content in rapeseed oil and its positive effects on lipids, particularly those mediated by oleic acid, which is the most abundant fatty acid in rapeseed oil. The literature around MUFAs is extensive, and we have tried to remain focused mainly on omega-3 fatty acids, as not to divert from the aim of our paper, also preventing our manuscript from becoming too lengthy.
The authors mention individual studies, but they do not try a meta-analysis. This agrees with the title (an overview), but one would whish to know the reason described why this has not been attempted.
Response: Our aim was to create a narrative review, as communicated with the handling editor who had invited us for the Special Issue. Most importantly however, our overview is a way to discuss the viability of various ALA-rich foodstuffs, instead of focusing on food item only, which would be achieved by a meta-analysis. The development of meta-analyses to draw conclusions over a particular item is of paramount importance, and for that reason we have added this message in the topic “Perspectives”
An important point in relation to fatty acid intake is the balance between n-6 and n-3 fatty acids. Could it be that some of the positive aspects ascribed to the ALA containing products could be due to reduced linoleic acid availability?
Response: We have further reinforced this rationale in the first paragraph of subtopic 6 “Decision-making practice”.
Specific points
Line 66: warrant instead of are warranted
Response: We have fixed this issue.
Line 92: LC-PUFA instead of PUFAs
Response: We have replaced PUFAs with long chain-PUFAs in that point.
Figure 1: the pathway for n-6 PUFA seems not correct
Response: We have checked and mended Figure 1. Thank you for bringing this issue to our attention.
Table 1: might be clearer to give the contents per 100g instead of per table spoon (13.6 g)
Response: We have chosen tablespoon as unit of measurement as it is a feasible way of translating nutrition facts to real-life, as 100 g/d of oil is not a very commonly used unit. The United States Department of Agriculture (USDA) database provides both values – /100g and tablespoon – and we feel this is a reasonable unit that can be easily translated into practice.
Line 302: an explanation would be good, how refinement could selectively eliminate ALA from rapeseed oil (considering that fatty acids are triglyceride bound)
Response: Triglyceride fatty acid composition of plant oils is influenced by the various chemical and physical processes of extraction from its seeds. Cold first press oils tend to contain triglycerides with lower melting point than oils obtained from hot and subsequent presses. Upon reflection, we have made the decision to remove the sentence where we had cited a possible loss of ALA under extraction and refinement processing, as it would divert the reader’s attention to a topic that is not the aim of our paper.
Reviewer 2 Report
In this manuscript (ID# nutrients-962861), titled “Beyond Fish Oil Supplementation: The Effects of Alternative Plant Sources of Omega-3 Polyunsaturated Fatty Acids upon Lipid Indexes and Cardiometabolic Biomarkers—An overview”, authors, Santos et al, summarized the effect of different plant sources of w3 polyunsaturated fatty acid (n3-PUFA) on lipid indexes and cardiovascular system using the data published previously. They conclude ALA supplementation, increased consumption of ALA-rich foodstuffs, combined with reduced omega-6 (n-6) PUFAs intake, could improve improve cardiometabolic and lipid profile. However, there are several major concerns, which are listed in the following paragraphs:
- The beneficial effect of n3-PUFAs on cardiovascular and lipid profile have been well studied. The conclusion of this review is not novel. Recommend: 1) Identify the gap in knowledge in this area, that need further investigation. Provide more information about the prospective direction in the future study; 2) Provide more discussion regarding the mechanisms underlying the effect of n3-PUFAs vs n6-PUFAs on cardiometabolic and lipid profile.
- The authors mentioned that plant source of n3 PUFAs may be better than the fish source could due to the other nutrients in the plant, such as fibers, minerals, and antioxidants, which are also benefit to the cardiometabolic and lipid profile. This information should be provided to support your conclusion.
Author Response
Reviewer 2
In this manuscript (ID# nutrients-962861), titled “Beyond Fish Oil Supplementation: The Effects of Alternative Plant Sources of Omega-3 Polyunsaturated Fatty Acids upon Lipid Indexes and Cardiometabolic Biomarkers—An overview”, authors, Santos et al, summarized the effect of different plant sources of w3 polyunsaturated fatty acid (n3-PUFA) on lipid indexes and cardiovascular system using the data published previously. They conclude ALA supplementation, increased consumption of ALA-rich foodstuffs, combined with reduced omega-6 (n-6) PUFAs intake, could improve improve cardiometabolic and lipid profile. However, there are several major concerns, which are listed in the following paragraphs:
Response: Dear reviewer, we express our gratitude for your considerations, which have improved our paper. We have incorporated your suggestions and mended the text accordingly.
- The beneficial effect of n3-PUFAs on cardiovascular and lipid profile have been well studied. The conclusion of this review is not novel. Recommend: 1) Identify the gap in knowledge in this area, that need further investigation. Provide more information about the prospective direction in the future study; 2) Provide more discussion regarding the mechanisms underlying the effect of n3-PUFAs vs n6-PUFAs on cardiometabolic and lipid profile.
Response: Regarding the gap in the area, we strongly believe that the major challenge around our topic is the actual translation of this knowledge into clinical settings. So much so that we have reemphasized in our introduction that there is a great interest in clarifying the clinical potential of plant sources of n3-PUFAs. In addition, we have created the subtopic “Perspectives” as a means of showing the gaps more evidently, and how further research in this field could assist.
Concerning the mechanisms of n3-PUFAs vs n6-PUFAs on cardiometabolic and lipid profile, we have added two subtopics “3.2. n3-PUFAs versus n6-PUFAs and 3.3. Cardiometabolic pathways), which we hope have further clarified the topics addressed.
- The authors mentioned that plant source of n3 PUFAs may be better than the fish source could due to the other nutrients in the plant, such as fibers, minerals, and antioxidants, which are also benefit to the cardiometabolic and lipid profile. This information should be provided to support your conclusion.
Response: After careful consideration and reappraisal of our paper, we do not feel that we have concluded that plant sources of n3 PUFAs may be better than the fish sources due to the other nutrients in the plant, such as fibre, minerals, and antioxidants. Following your suggestion however, we have reemphasized the importance of fibre, minerals (potassium and magnesium) and non-essential substances in the first paragraph of the topic “Conclusions”.
Reviewer 3 Report
The review by Santos et al "Beyond Fish Oil Supplementation: The Effects of Alternative Plant Sources of Omega-3 Polyunsaturated Fatty Acids upon Lipid Indexes and Cardiometabolic Biomarkers - An overview" provides an up-to-date overview of randomized controlled trial results on the efficacy of plant-derived alpha-linolenic acid from foodstuffs and supplements upon lipid profile and other cardiometabolic markers. However, there are some concerns.
The authors considered some plant foods or edible oils rich in alpha-linolenic acid (ALA) and discussed studies demonstrating their beneficial effects. However, some of these sources are richer in linoleic acid (LA) than ALA (hemp seeds, nuts, rapeseed oil, and soybeans). Several pieces of evidence show a positive effect of ALA on cardiometabolic biomarkers (see for example [1], cardiovascular disease, and mortality (see for example [2]) and diabetic risk (see for example [3]). Besides, as the authors themselves also say, ALA is not sufficiently transformed into EPA, and DHA in humans and the plant foods considered are also rich in many other compounds that have many cardiometabolic effects similar to EPA and DHA on metabolic biomarkers. For all these reasons, not only is both the review approach and the conclusions drawn by the authors highly questionable, but they could also provide a misleading message about ALA properties.
- Ramsden, C.E.; Zamora, D.; Majchrzak-Hong, S.; Faurot, K.R.; Broste, S.K.; Frantz, R.P.; Davis, J.M.; Ringel, A.; Suchindran, C.M.; Hibbeln, J.R. Re-evaluation of the traditional diet-heart hypothesis: Analysis of Recovered data from Minnesota Coronary Experiment (1968-73). BMJ 2016, 353.
- Marklund, M.; Wu, J.H.Y.; Imamura, F.; Del Gobbo, L.C.; Fretts, A.; De Goede, J.; Shi, P.; Tintle, N.; Wennberg, M.; Aslibekyan, S.; et al. Biomarkers of Dietary Omega-6 Fatty Acids and Incident Cardiovascular Disease and Mortality: An Individual-Level Pooled Analysis of 30 Cohort Studies. Circulation 2019, 139, 2422–2436.
- Wu, J.H.Y.; Marklund, M.; Imamura, F.; Tintle, N.; Ardisson Korat, A. V.; de Goede, J.; Zhou, X.; Yang, W.S.; de Oliveira Otto, M.C.; Kröger, J.; et al. Omega-6 fatty acid biomarkers and incident type 2 diabetes: pooled analysis of individual-level data for 39 740 adults from 20 prospective cohort studies. Lancet Diabetes Endocrinol. 2017, 5, 965–974.
Author Response
Reviewer 3
The review by Santos et al "Beyond Fish Oil Supplementation: The Effects of Alternative Plant Sources of Omega-3 Polyunsaturated Fatty Acids upon Lipid Indexes and Cardiometabolic Biomarkers - An overview" provides an up-to-date overview of randomized controlled trial results on the efficacy of plant-derived alpha-linolenic acid from foodstuffs and supplements upon lipid profile and other cardiometabolic markers. However, there are some concerns.
The authors considered some plant foods or edible oils rich in alpha-linolenic acid (ALA) and discussed studies demonstrating their beneficial effects. However, some of these sources are richer in linoleic acid (LA) than ALA (hemp seeds, nuts, rapeseed oil, and soybeans). Several pieces of evidence show a positive effect of ALA on cardiometabolic biomarkers (see for example [1], cardiovascular disease, and mortality (see for example [2]) and diabetic risk (see for example [3]). Besides, as the authors themselves also say, ALA is not sufficiently transformed into EPA, and DHA in humans and the plant foods considered are also rich in many other compounds that have many cardiometabolic effects similar to EPA and DHA on metabolic biomarkers. For all these reasons, not only is both the review approach and the conclusions drawn by the authors highly questionable, but they could also provide a misleading message about ALA properties.
- Ramsden, C.E.; Zamora, D.; Majchrzak-Hong, S.; Faurot, K.R.; Broste, S.K.; Frantz, R.P.; Davis, J.M.; Ringel, A.; Suchindran, C.M.; Hibbeln, J.R. Re-evaluation of the traditional diet-heart hypothesis: Analysis of Recovered data from Minnesota Coronary Experiment (1968-73). BMJ 2016, 353.
- Marklund, M.; Wu, J.H.Y.; Imamura, F.; Del Gobbo, L.C.; Fretts, A.; De Goede, J.; Shi, P.; Tintle, N.; Wennberg, M.; Aslibekyan, S.; et al. Biomarkers of Dietary Omega-6 Fatty Acids and Incident Cardiovascular Disease and Mortality: An Individual-Level Pooled Analysis of 30 Cohort Studies. Circulation 2019, 139, 2422–2436.
- Wu, J.H.Y.; Marklund, M.; Imamura, F.; Tintle, N.; Ardisson Korat, A. V.; de Goede, J.; Zhou, X.; Yang, W.S.; de Oliveira Otto, M.C.; Kröger, J.; et al. Omega-6 fatty acid biomarkers and incident type 2 diabetes: pooled analysis of individual-level data for 39 740 adults from 20 prospective cohort studies. Lancet Diabetes Endocrinol. 2017, 5, 965–974.
Response: Dear reviewer, thank you very much for your comments and references. We agree that the effects of plant nutrients are beyond ALA only, and other components do indeed play major roles in overall metabolic modulation. We have reinforced more clearly the synergistic effects of ALA alongside fibre, minerals and non-essential substances, including sterols and polyphenols, in the first paragraph of the topic “Conclusions”.
We have revisited and further emphasized the importance of other nutrients and compounds found in ALA-rich foods, presenting not only their ALA content but also the nutrition facts of minerals with crucial relevance in cardiovascular health (e.g. calcium, magnesium, and potassium), total lipids, MUFAs, PUFAs, n−6/ n−3 ratio, and total fibre, presented in Tables 1 and 2.
We appreciate the references suggested. Their evident quality can be noted due to the journals in which they were published, sample size, biomarkers used and medical outcomes. Correspondingly, we have added a new background point in the second paragraph of ‘Introduction’ in order to appropriately refer to them in our paper.
Round 2
Reviewer 1 Report
Thanks to the authors for considering the suggestions.
There are no further questions from my side.
There are two minor Points from my side:
line 169: "reducing VLDL-C Synthesis", I think it would be better to use VLDL Synthesis, as this does not influence cholesterol synthsis
Line 607: in the title of this reference should it be 13C-U-eicosapentaenoic acid
Reviewer 2 Report
The manuscript has been improved. No further concern.
Reviewer 3 Report
The manuscript has been revised and the previously reported criticisms have been resolved